# Developing and Disseminating Physical Activity Messages Targeting Parents: A Systematic Scoping Review

**DOI:** 10.3390/ijerph18137046

**Published:** 2021-07-01

**Authors:** Victoria Larocca, Kelly P. Arbour-Nicitopoulos, Jennifer R. Tomasone, Amy E. Latimer-Cheung, Rebecca L. Bassett-Gunter

**Affiliations:** 1School of Kinesiology & Health Science, York University, Toronto, ON M3J 1P3, Canada; rgunter@yorku.ca; 2Faculty of Kinesiology & Physical Edication, University of Toronto, Toronto, ON L5L 1C6, Canada; kelly.arbour@utoronto.ca; 3School of Kinesiology and Health Studies, Queen’s University, Kingston, ON K7L 3N6, Canada; tomasone@queensu.ca (J.R.T.); amy.latimer@queensu.ca (A.E.L.-C.)

**Keywords:** physical activity, health promotion, children, children with disabilities, messaging

## Abstract

Background: Physical activity (PA) messages have demonstrated success in targeting parent support for PA. However, little research exists to inform the development and dissemination of optimally effective PA messages targeting parents. A synthesis of existing literature is necessary to inform message development and dissemination strategies. Unique considerations for parents of children with disabilities (CWD) should be identified given a need for inclusive PA messaging that consider the needs of CWD and their families. Methods: Systematic scoping methodologies included a peer-reviewed literature search and expert consultation to identify literature regarding PA messages targeting parents, and considerations for parents of CWD. Results: Thirty-four articles that met eligibility criteria were included for examination. Twenty-eight studies were identified regarding the PA messages targeting parents; six themes and 12 subthemes emerged from these articles. Six studies were identified regarding unique considerations for parents of CWD; three themes and four subthemes emerged from these articles. Conclusions: Through knowledge synthesis, this research can contribute to a knowledge translation process to inform practice guidelines for the development and dissemination of PA messages targeting parents, while also providing unique considerations for PA messages targeting parents of CWD.

## 1. Introduction

Physical activity among children is on the decline. For example, only one third of children in Canada and one quarter of children in the United States are meeting recommended physical activity (PA) guidelines [1,2]. There is a need for strategies to increase PA among children and it is suggested that such strategies should target parent support for PA [3,4,5] as it is an important determinant of PA participation among children [5,6,7,8,9].

One strategy for increasing parent support for PA is through the development and dissemination of targeted PA messages [10]. Messages can include motivational statements or information about PA [11], such as the benefits of PA or the importance of supporting PA. Messages can be disseminated (i.e., purposely distributed) via different media channels (e.g., social media campaigns, mass media messaging) to a target audience such as parents [12]. Various PA campaigns have demonstrated success using persuasive messages targeting parent support for PA and its antecedents (e.g., motivation, attitudes, self-efficacy, planning) [10,13,14,15,16]. However, there are no known practice guidelines for the development and dissemination of PA messages targeting parents. An important first step in understanding such practices is a comprehensive synthesis of the peer-reviewed literature to identify strategies that have been employed in research regarding PA messages targeting parents.

When developing and disseminating PA messages targeting parents, it may be necessary to consider the unique needs of parents of children with disabilities (CWD) [17]. CWD are less active than their peers without disabilities [18] and often rely on parent support to facilitate PA [8,19]. PA promotion efforts have been successful in increasing planning to provide support for PA [20] and psychosocial antecedents of parent and self-reported parent support for PA [21] holding promise as a strategy to motivate support for PA among parents of CWD [17,22]. Parents of CWD have expressed unique and specific messaging needs [17] and there has been a call for inclusive PA messaging practices that meet the specific needs of people with disabilities [23]. However, there is currently no known synthesis of research regarding strategies for developing and disseminating PA messages targeting parents of CWD. Considering the unique needs of parents of CWD could inform practice guidelines for motivating PA support among parents of CWD through targeted messaging.

The Framework for Knowledge Transfer [24] can guide the process of consolidating research findings to aid informed decision making around message development and dissemination to a target audience. Within the context of the current review, the framework guided the clear identification of (a) a target audience (i.e., parents and parents of CWD), (b) literature supporting the development of PA messages targeting parents and (c) literature supporting the dissemination of PA messages targeting parents. Therefore, this review has two purposes: (a) to identify considerations within the peer-reviewed literature regarding the development and dissemination of PA messages targeting parents and (b) to identify considerations regarding the development and dissemination of PA messages specifically targeting parents of CWD.

## 2. Materials and Methods

A systematic scoping methodology was used to couple the rigor and replicability of a systematic review with the exploratory lens of a scoping review [25]. This review was conducted according to the PRISMA Extension for Scoping Reviews (PRISMA-ScR) Checklist [26] and the following systematic scoping methods [25,27,28,29]: (a) identifying the research questions, (b) identifying relevant search records, (c) record selection, (d) charting the data, and (e) collating, summarizing, and reporting results.

### 2.1. Identifying the Relevant Search Records

A peer-reviewed literature search and expert consultation were used to capture broad and comprehensive literature [29]. Search terms were determined in consultation with a librarian after the detailed assessment of indexing terms applied to a ‘known’ set of articles [10,17,30,31]. A combination of terms for PA, exercise, parents, child, mass media, dissemination, and messages was applied for the searches to identify relevant literature. In addition to these terms, the term disability was added in a secondary search to identify research specific to parents of CWD. An example of a full electronic search strategy is as follows: A keyword search string of “physical activity AND parents AND child AND messages OR mass media OR dissemination” was entered into the journal database PsychInfo. Limits were applied in line with eligibility criteria. The following concepts were defined before conducting the search to provide clarity and consistency for the researchers: PA, parent, child, disability, messages, and dissemination (See Table 1).

### 2.2. Eligibility Criteria

Articles were limited to peer-reviewed publications. Inclusion criteria included: (a) a target population of parents or parents of CWD, (b) the age of the children (if specified) up to and including 24 years of age because UNESCO recognizes anyone up to the age of 24 a child or youth, (c) published between 2000–2020, and (d) full text available in English. Along with meeting the required eligibility criteria above, the focus of the article must have also included one of more of the following: (a) PA messages, (b) PA message and development strategies, (c) dissemination strategies. Exclusion criteria included: (a) a target population that did not include parents, (b) the age of the children is over 24 years of age, (c) published outside of 2000–2020, and (d) full text not available in English. Articles were excluded if they discussed the promotion of other health behaviours but not specifically PA or if they discussed caregivers but not specifically parents. Articles were included if they discussed multiple health behaviours with specific discussion of PA, although only information regarding PA was examined for the review. Likewise, articles were included if parents were discussed alongside caregivers more broadly.

### 2.3. Peer-Reviewed Databases

A systematic scoping search of the following databases was performed by a single researcher (May–June 2019): (a) CINAHL, (b) PsychInfo, (c) PubMed, (d) Scopus, (e) Sport Discus, and (f) Google Scholar. The first search identified literature regarding PA message development and dissemination targeting parents. The second search identified literature regarding PA message development and dissemination specifically targeting parents of CWD. A total of eight search term combinations using traditional “AND OR” approaches were applied to all six databases. Google Scholar searches included only the first five pages of each search result [32]. Searches were replicated on 1 December 2020 to ensure that papers published since the initial search were included.

### 2.4. Expert Consultations 

Content experts (*n* = 28) included the first author listed on each of the records identified through database searches. They were contacted via email (September–October 2019) and informed of the study objectives and were asked to provide any relevant literature. Four authors could not be reached, 18 authors responded, six authors did not respond.

### 2.5. Record Selection and Charting the Data

A reference managing software (i.e., Mendeley, London, United Kingdom) was used. The initial title and abstract screening was performed by one researcher (initials removed for blind submission). Two researchers (initials removed for blind submission) screened the full-text of remaining records independently based on the eligibility criteria. Among these two researchers, there was an 80.5% agree when screening full-texts for inclusion. Any discrepancies in agreement were resolved by discussion and consensus amongst all authors. One researcher (initials removed for blind submission) manually searched the reference lists of eligible articles to identify any additional relevant records and screened records from the expert consultations. A second researcher audited the manual search. 

### 2.6. Collating, Summarizing, and Reporting Results

One researcher (initials removed for blind submission) extracted the following data from eligible articles: (a) record characteristics (i.e., author, title, year, study design, and participant characteristics), (b) article focus (i.e., message development, message dissemination, or both), (c) message development or dissemination strategy used or discussed, and (d) key findings. One researcher developed preliminary themes and subthemes across the records using a thematic analysis approach. These themes and subthemes were discussed and finalized among all authors.

## 3. Results

### 3.1. Search Results Regarding Parents

A total of 2708 records were identified for screening (*n* = 2703 from database searches, *n* = 2 from expert consultations, *n* = 3 from hand searching reference lists). Following removal of duplicates and records that did not meet inclusion criteria (*n* = 2635), 73 studies remained. After reviewing these records according to inclusion criteria, twenty-eight studies were included regarding the primary purpose of the review (See Figure 1). 

#### 3.1.1. Evidence Characteristics 

Of the 28 studies included for review, 25 were empirical and included nine experimental studies (five pre-post, two post-test only, one between groups, one single time point) and 16 non-experimental studies (10 qualitative, four cross-sectional, and two descriptive studies). The 25 empirical studies took place in the following locations: Canada (*n* = 12), United States (*n* = 7), Australia (*n* = 5), and UK (*n* = 1). Three studies were non-empirical descriptive studies and took place in the United States (*n* = 2) and Chile (*n* = 1). Participants in these studies were mostly mothers with children between the ages of two and 17 years old.

#### 3.1.2. Summary of Main Findings: Development of PA Messages Targeting Parents

Three themes and six subthemes emerged regarding the development of PA messages targeting all parents. Relevant findings are presented in Table 2.

##### Theme 1: Message Persuasion 

(a) Message Framing. Two studies discussed framing as a strategy for developing PA messages [15,30]. The first study found that gain-framed messages were more effective compared to loss-framed messages among parents [15]. Alternatively, the second study found no differences in effectiveness between gain- and loss-framed messages among parents [30]. 

##### Theme 2: Messages That Consider Barriers to Parent Support for PA

(a) Common Barriers to Parent Support for PA. Four studies discussed common barriers to parent support for PA: time [34,35,36,37], safety concerns and lack of facilities [35], money [34,35], weather [35,36,37], and parents’ motivation [36]. 

(b) Information Regarding PA Guidelines. Three studies focused on messaging regarding PA guidelines as something to consider when developing PA messages targeting parents [34,37,38]. Parents’ lack of awareness or poor understanding of PA guidelines was highlighted [38]. All three studies emphasized that PA guidelines are a source of confusion for some parents [34,37,38]. Parents’ need for clarification surrounding the various types of PA (i.e., light, moderate, or vigorous) and their confusion regarding how to monitor their children’s PA [34] were discussed. However, when parents were aware of PA guidelines and had a good understanding, the guidelines were positively accepted [37].

(c) Guilt and Stress. Four studies identified that PA messages can evoke feelings of guilt and stress among parents [34,37,38,39]. Feelings of guilt and stress should be considered since PA can be perceived as ‘something else to worry about’ by parents [34,37]. Feelings of guilt and stress regarding monitoring children’s PA behaviour were common among parents [34,37,38,39] and were not perceived as motivating [39].

##### Theme 3: Messages That Target Parents’ Attitudes 

(a) Attitudes Toward Child PA. Eight studies discussed targeting attitudes toward child PA as a message development strategy [14,33,35,36,38,40,42]. Parents generally hold positive attitudes toward child PA [14,35,36,38] and PA messages targeting parents were found to have positive effects on attitudes toward child PA [33,40,42]. However, many parents expressed that increasing PA for their own children was not an issue and believed that their own children were meeting PA guidelines [14,38,41]. This perception has been observed among parents of children who are not meeting PA recommendations [14,38].

(b) Attitudes Toward Parent Support for PA. Two studies discussed targeting attitudes toward parent support for PA as a message development strategy [42,43]. These studies suggest that targeting parents’ affective attitudes through providing information about the benefits of parent support for PA can be an effective strategy to motivate parents [42,43].

##### Dissemination of PA Messages Targeting Parents

Three themes and six subthemes emerged regarding the dissemination of PA messages targeting parents. Main relevant findings are presented in Table 3.

##### Theme 1: Dissemination to Enhance Cognitive Processing 

(a) Awareness. Three studies discussed message dissemination strategies to enhance awareness which was considered as an important factor in evaluating PA message dissemination effectiveness among parents [10,42,44]. Further, all three studies highlighted the importance of garnering awareness to promote changes in beliefs and intentions toward child PA and parent support for PA [10,42,44]. In these studies, message awareness was positively associated with parents’ favourable attitudes toward child PA [42], believing the benefits of child PA [44], believing their child needs to engage in more PA, having stronger intentions, and demonstrating more support for PA [10].

(b) Recall. Four studies discussed message dissemination strategies to enhance recall as a strategy to strengthen dissemination effectiveness among parents [42,44,45,46]. PA message recall among parents was positively associated with attitudes, beliefs, and support for PA behaviours [42], as well as greater knowledge regarding child PA and increased family PA [45]. However, PA message recall was generally low among parents44 unless it was prompted [46].

##### Theme 2: Social Marketing Strategies to Enhance Dissemination

(a) Marketing strategies. Three studies highlighted the success of PA campaigns that targeted parents through various social marketing strategies such as [47,48,49] extensive consumer research [47,48], appealing branding [48] and brand affinity (i.e., messaging that aligned with the parents’ values). Some marketing strategies related to dissemination included paid media advertising, contests and community events, and collaborations to promote parent support for PA [49].

(b) Tailoring dissemination for subgroups of parents. Two studies discussed the use of tailoring to optimize dissemination of messages to subgroups of parents [48,49]. Tailored dissemination approaches such as messages in different languages or disseminated by ethnic organizations enhanced parents’ attention to messages, perceptions of relevance and subsequent motivation [48] which was successful in promoting parent support for PA [48,49].

(c) Brand Equity. Two studies discussed the use of brand equity as dissemination strategy [15,47]. The first study used popular television characters as a strategy to disseminate PA messages. The strategy led to increased brand equity over a two-year period [47]. The second study highlighted the importance of promoting feelings of credibility and loyalty which increased parents’ brand equity after six months and parent support for PA [15].

##### Theme 3: Messages That Target the Dissemination Preferences and Suggestions of Parents

(a) Preferred message dissemination approaches. Seven studies discussed parents’ preferred PA message dissemination approaches which included: multi-platform dissemination approaches [34,50,52] including both digital (i.e., social media, apps, websites) and traditional (i.e., brochures, magazines, television, or radio) forms of dissemination [34,53], text messaging [54,55], and unique forms dissemination (e.g., storybooks, community “parent nights”) [50,51]. Regardless of dissemination method, parents prefer to receive messages from credible sources such as community centres, public health institutions, schools, doctors, or government agencies [34].

### 3.2. Search Results Regarding Parents of CWD

A total of 2059 records were identified for screening (*n* = 2057 from database searches, *n* = 2 from expert consultations, *n* = 0 from hand searching). Following removal of duplicates and records that did not meet inclusion criteria (*n* = 2024), 35 studies remained. After reviewing these records according to inclusion criteria, six studies were included regarding the secondary purpose of the review (See Figure 2).

#### 3.2.1. Evidence Characteristics 

All six studies were empirical and were non-experimental designs (five qualitative and one content analysis) and took place in Canada (*n* = 5) and the UK (*n* = 1). Participants were parents with at least one CWD (age 5–21 years).

#### 3.2.2. Summary of Main Findings: Development of Physical Activity Messages Targeting Parents of CWD

Two themes and three subthemes emerged regarding the development of PA messages targeting parents of CWD. Main relevant findings are presented in Table 4. 

##### Theme 1: Common Barriers to Parent Support for PA among Parents of CWD

(a) Lack of Information. Six studies discussed a lack of PA messages targeted toward parents of CWD as a barrier to providing parent support for PA as well as an important consideration for message development targeting these parents [17,56,57,58,59,60]. The lack of targeted PA information resulted in feelings of frustration among parents of CWD [56,57,59]. The following types of PA information were identified as specifically lacking and desired by parents of CWD: (a) clear and consistent definitions of terms used to describe PA programming (e.g., accessible, inclusive, adapted) [34,59], (b) details regarding the accessibility and inclusivity of PA activities [57,59], (c) information regarding safety considerations for specific PA activities [17], (d) information regarding support options available with programs [58,59], and e) ideas for facilitating PA at home through tools such as checklists or choice boards [17].

(b) PA Barriers Salient to Parents of CWD. Five studies discussed common barriers salient to parents of CWD as important to consider such as PA costs, time constraints, lack of accessibility or support [57], lack of disability-specific PA opportunities [58], lack of targeted PA information [17,57,59], social inclusion [56], and safety [56,57,59]. Three studies highlighted the extraordinary efforts that parents of CWD reported to overcome barriers to supporting their children’s PA [17,56,59]. The following messaging approaches were suggested: (a) provide information regarding coaches or program staff training [59], (b) use inclusive images [17,57], and (c) include information regarding the safety of PA programs and opportunities [56,57,59].

##### Theme 2: Messages That Target Psychosocial Antecedents of Parent Support for PA Among Parents of CWD

(a) Theoretical Predictors of Behaviour Change. One study discussed the need for theoretically based PA messages targeting parents of CWD as a consideration for message development targeting parents of CWD [60]. For example, less than 10% of the PA website content targeting parents of CWD included messages targeting self-regulation, self-monitoring, and planning [60].

##### Dissemination of Physical Activity Messages Targeting Parents of CWD

One theme and one subtheme emerged regarding the dissemination of PA messages targeting parents of CWD. Main relevant findings are presented in Table 5.

##### Theme 1: Dissemination Preferences and Suggestions of Parents of CWD

(a) Preferred Dissemination Approaches. Three studies discussed the PA message dissemination preferences of parents of CWD which include [17,58,59]: a multi-platform approach to PA message dissemination from credible and reliable sources [17,58,59] that includes a “central hub” for PA information [17]. Preferred dissemination channels include blogs, chat rooms and message boards to find and share PA information with other parents of CWD [17].

## 4. Discussion

This is the first known systematic scoping review to identify peer-reviewed literature regarding the development and dissemination of PA messages for parents, while also identifying additional consideration for targeting parents of CWD.

### 4.1. Strategies Regarding the Development of Physical Activity Messages

The Framework for Knowledge Transfer [24] suggests that organizations should develop information from a breadth of research to provide enhanced validity and justification for message content development. Therefore, when developing PA messages for parents, it is important to consider the results of multiple studies to make an evidence-informed decision regarding the development of PA message content. Despite the mixed findings regarding message framing [17,33], it is cautiously recommended that PA messages targeting parents should be gain-framed. Gain-framed messages have been considered optimal for promoting health prevention behaviours such as PA among varying populations61 and among parents [22] as well as proxy behaviours to support PA (e.g., parent support) [61]. Gain-framed PA messages are often rated more favourably than loss-framed [11,62] and should maintain focus on what parents can do to encourage and support PA for their child [38,63,64].

Although parents generally rank PA as a high priority for their children [5], they often encounter barriers to providing support for PA [34,35,36,37]. Taking identified barriers into consideration [34,35,36,37], providing information regarding common barriers may serve to enhance parents’ self-efficacy or perceived behavioural control regarding their abilities to provide support for PA. Previous messaging campaigns targeting parents have been effective in addressing common barriers to support for PA [36,47,49]. While some barriers (e.g., financial limitations, weather) may be difficult to address via messages alone, providing parents with informational tools or examples may help to manage barriers [30] (e.g., ideas for free, indoor, or low-cost PA).

Another barrier identified for this review was information regarding PA guidelines. Although many parents value PA guidelines [37], unclear guidelines can act as a source of confusion [34,37,38]. Parents desire clear PA guidelines, consistent definitions, examples of different types of PA at various intensities, and suggestions for assessing and monitoring a child’s PA levels [34]. Organizations wishing to target parents should take these needs into consideration and include some of this desired information when developing PA messages. PA messages should be supportive, positive and pragmatic to evoke feelings of motivation and achievement [34,38] rather than messages that make parents feel guilty for their child’s inactivity, which are demotivating [39]. Similar findings have been observed among parents regarding other behaviour guidelines (e.g., sedentary behaviour) [64]. Using messages to address barriers to supporting child PA can boost parents’ motivation through enhanced perceived behavioural control, which has been identified as the strongest predictor of child PA [5,65].

Messaging strategies that target parents’ attitudes may help parents recognize the crucial role that PA plays in child development and well-being [52] and these messages can serve as a booster to motivate parent support for PA [30]. Messages should also specifically target attitudes toward parent support for PA as it is the strongest predictor of parents’ motivation [5]. Given that affective attitudes impact parents’ intentions to provide support for PA over time, PA messages should pay particular attention to the affective benefits of parent support for PA in addition to its instrumental benefits [5].

### 4.2. Unique Message Development Considerations for Parents of CWD

Although many message development strategies may be universal to all parents, barriers that are salient to parents of CWD should be considered. A lack of targeted PA messages is a significant barrier to parent support for PA among parents of CWD [17,56,57,58,59,60]. There has been a call for inclusive and targeted approaches to PA message development targeting those within the disability community [23]. Effective PA messages must meet the needs of a target audience (i.e., parents of CWD) [17] and targeted PA messages are desired by parents of CWD [17,57,59]. There is great value in providing disability-specific PA information to parents of CWD [66] as it can help reduce their perceptions of barriers and enhance self-efficacy to provide support for PA [17].

PA messages targeting people with disabilities tend to lack theory-based content [66] and it is recommended that PA messages targeting parents of CWD include content regarding theoretical predictors of behaviour change to further enhance message effectiveness [11]. For example, there may be value in developing messages that target the important predictors of parent support for PA among parents of CWD such as perceived behavioural control, subjective norms, attitudes toward parent support for PA, motivation, behavioural regulation and planning [67,68,69,70]. Parents of CWD have specifically expressed a desire for messages regarding tools and strategies to support their own planning and self-regulatory behaviours [17].

### 4.3. Strategies Regarding the Dissemination of Physical Activity Messages

The Framework for Knowledge Transfer [24] highlights the importance of using evidence-informed message dissemination efforts as well as engaging researchers and community-based organizations to identify optimally effective message dissemination strategies. PA messages targeting parents should be disseminated in a way that maximizes cognitive processing pathways (i.e., awareness and recall). While awareness and recall do not necessarily directly translate into changes in parent support for PA [10,42,45], PA messages that utilize dissemination strategies to effectively evoke awareness and recall may positively affect antecedents of behaviour such as attitudes and perceived behavioural control [13,22,71]. While more research is needed to understand practices to garner awareness and recall, one suggested dissemination strategy is the use of consistent and repeated exposure of PA messages [14].

Social marketing strategies such as message tailoring, brand equity, channel placement, and outcome evaluations [71,72,73] may optimize dissemination efforts [74,75]. While these marketing strategies have only been used within a small number of PA campaigns and among varying populations, they hold promise as an effective strategy to enhance the dissemination of PA messages to parents. One social marketing strategy includes tailoring dissemination strategies to subgroups of parents to meet individual-level needs and preferences [76]. Some examples include using media channels that are preferred by the target audience (e.g., mothers versus fathers, various age groups, city of residence) [76] or targeting parents of CWD [17]. Tailored dissemination strategies are thought to be more effective compared to generic or non-tailored messages [76] as they can improve parents’ overall uptake of PA messages by enhancing attention, perceptions of relevance and subsequent motivation to provide support for PA [48]. Brand equity is another social marketing strategy which refers to the perceived value that a brand brings to a consumer [33]. Brand equity is an important component of PA message dissemination [77,78] and has been linked to increased parent support for PA [42,47]. PA messaging campaigns targeting parents may benefit from investing resources to build brand equity [41] which can be done by disseminating messages through sources that assure credibility and loyalty.

Parents prefer a combination of digital and traditional means of dissemination [34,53] to enhance message effectiveness [24]. Credible sources such as community centres, public health institutions, schools, doctors, or government agencies should disseminate PA messages [22,34] through various channels and times [79]. A multi-phase and integrated social marketing campaign is beneficial when disseminating PA messages to a target audience [80].

### 4.4. Unique Message Dissemination Considerations for Parents of CWD

Many dissemination strategies may be universally effective. However, parents of CWD have expressed a need to feel understood as an audience [58]. Dissemination strategies that meet the needs and preferences of parents of CWD as an audience can optimize information seeking behaviours [81] and message effectiveness [24] while also enhancing important antecedents to behaviour change such as awareness and recall [13,70]. Consideration for the unique preferences identified in this review (e.g., multi-platform approach, “central hub” for information, blogs, and chatrooms) are recommended when disseminating PA messages targeting parents of CWD via credible sources.

### 4.5. Strength and Limitations

This systematic scoping review is the first of its kind aimed at identifying strategies to inform the development and dissemination of PA messages targeting parents, while also identifying unique considerations for targeting parents of CWD. The results of this review contribute to the general PA messaging literature as well as informing PA message development and dissemination strategies that are inclusive of the needs of parents of CWD. Although the methodological rigor [25,26] is a strength of the review, there are several limitations that must be acknowledged. First, there was no assessment of the quality of the articles selected for the review which could lead to a risk of bias but this approach allowed for a breadth of the research to be included. Further, the purpose of this review was to identify literature regarding the development and dissemination of PA messages targeting parents. However, the messaging and dissemination strategies incorporated in the literature were not evaluated for effectiveness in this study. Evaluating the effectiveness of PA messages among parents and parents of CWD is an important next step to understand how to best develop and disseminate messages to these populations. Further, to improve the quality of future reviews on this topic, a criterion for the number of cited references should be considered when developing the results. Second, the inclusion criteria resulted in the selection of a relatively small number of articles. A broader inclusion criteria could retrieve articles from a multi-disciplinary perspective (e.g., broader fields of messaging, social marketing, other health behaviour promotion) that highlight other strategies and considerations that could be applied to the development and dissemination of PA messages targeting parents. For example, research regarding the use of messages to promote other health behaviours may be useful in informing the development and dissemination of messages targeting parent support for PA. This may also help with yielding more studies to meet criterion for the number of cited references.

## 5. Conclusions

The results of this systematic scoping review have pragmatic implications in informing PA message development and dissemination in practical settings, and can inform the development of practice guidelines for creating and disseminating PA messages targeting all parents and parents of CWD. Future research is encouraged to explore message development and dissemination strategies from other areas of health promotion and social marketing to further inform how to optimally develop and disseminate PA messages to parents. Further, it is suggested that future research also consider various mechanisms of health behaviour change in order to move messaging research forward and understand which mechanisms might be best to target through the use of persuasive PA messages. The information synthesized within this review can be used to guide future research as well as PA organizations wishing to develop and disseminate messages to parents to motivate parent support for PA. The review provides unique considerations for the development and dissemination of PA messages targeting parents of CWD, which are important in addressing the call for inclusive PA messaging [23]. The development and dissemination of evidence-informed PA messages targeting parents can optimize their impact in motivating parent support for PA and ultimately PA participation among all children.

## Figures and Tables

**Figure 1 ijerph-18-07046-f001:**
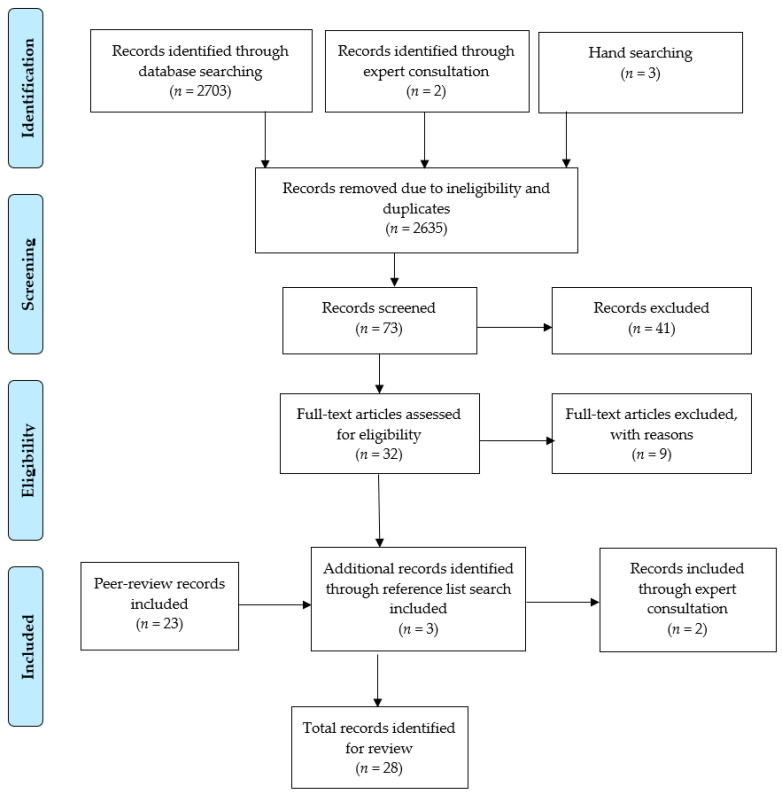
PRISMA study selection flow chart regarding parents.

**Figure 2 ijerph-18-07046-f002:**
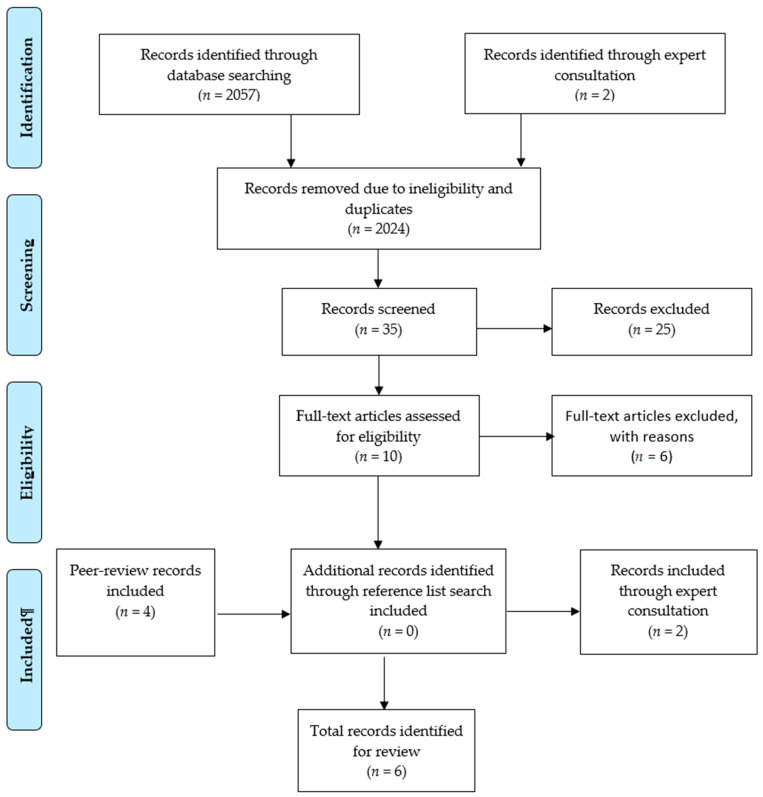
PRISMA study selection flow chart regarding parents of CWD.

**Table 1 ijerph-18-07046-t001:** Working definitions for the purpose of this research.

Term	Working Definition
Physical Activity (PA)	Any bodily movement produced by skeletal muscles that results in energy expenditure and results in increased heart rate and breathing was used to describe both structured PA such as sports and programs, as well as leisure time unstructured PA such playing with friends, dancing, or walking. Active transportation was also included. Types of “play” were included in the review as long as they were specified as physical or active play.
Parent	Biological or legal guardian and/or caregiver.
Child	Anyone up to and including age 24.
Disability	Activity limitation or participation restrictions caused by physical or cognitive impairment.
Messages and information	All information or knowledge about PA to be conveyed to a message recipient. All forms of information and messages were allowable and included (e.g., digital, print, radio).
Dissemination	Distribution of messages and information to a target audience via purposeful channels and strategies. All forms of dissemination were allowable and included (e.g., social media campaigns, mass media messaging/commercials, posting guidelines on websites, or communications with a practitioner).

**Table 2 ijerph-18-07046-t002:** Literature regarding the development of physical activity messages targeting parents.

Theme	Subtheme	Articles Identified	Main Relevant Findings	Recommendation for PA Message Development
(1) Message persuasion	(a) Message framing	[30,33]	Gain-framed PA messages targeting parents were more effective in promoting message engagement, believability, positive attitudes, and overall favourability compared to loss-framed messages [33]. Gain- and loss-framed PA messages were equally effective [30].	Messages targeting parents should be gain-framed to promote motivation and encouragement to provide support for PA.
(2) Messages that consider barriers to parent support for PA	(a) Common barriers to parent support for PA	[34,35,36,37]	Common barriers to providing parent support for PA include time [34,35,36,37], safety concerns [35], money [34,35] weather [35,36,37], lack of facilities [35], and parents’ motivation to provide support for PA [37].	Messages targeting parents should address common barriers they experience (e.g., time, money, safety, and weather). Such messages can boost parents’ perceived control over providing support for PA by enhancing their self-efficacy.
	(b) Information regarding PA guidelines	[34,37,38]	Many parents are unaware of PA guidelines, or exhibit low knowledge of PA guidelines [38]. To enhance understanding, parents desire clarity around definitions of PA [38], classifications of PA intensities (e.g., light, moderate, vigorous) [34] and examples of different types of PA to alleviate some confusion surrounding PA guidelines [37].	Messages targeting parents should provide parents with practical information regarding PA guidelines. For example, providing parents with examples of different types of PA (e.g., light, moderate and vigorous), providing clear definitions of what qualifies as PA, and strategies to assess their child’s PA.
	(c) Guilt and stress	[34,37,38,39]	Persuasive PA messages can evoke feelings of stress among parents [38] as PA guidelines are perceived as something else to worry about by parents [34,37]. Parents were not motivated by messages that evoked feelings of guilt and rather these messages were negatively associated with parents’ perceived behavioural control and intentions to provide support for PA [39].	Messages targeting parents should remain supportive, positive and pragmatic in order to provide parents with feelings of motivation and achievement rather than promoting feelings of guilt and stress.
(3) Messages that target parents’ attitudes	(a) Attitudes toward child PA	[14,15,35,36,38,40,41,42]	The existing research suggests that parents generally hold positive attitudes and perceptions toward child PA [14,35,36,38]. Parents who felt motivated after viewing a PA message had more positive attitudes toward child PA [40]. Positive effects of messages on parents’ attitudes toward child PA have been observed [15,42]. A common disconnect between parents’ attitudes toward child PA in general and their *own* child’s PA [14,38,41] was identified.	Messages targeting parents should focus on the presenting the benefits of child PA and present strategies to help increase their own child’s PA. Such messages may serve as a booster to parent support for PA.
	(b) Attitudes toward parent support for PA	[42,43]	The current literature suggests that targeting parents’ affective attitudes toward support for PA can be an effective strategy to motivate parents to provide support [43]. PA campaigns targeting parents have had success in motivating parents to provide support by providing them with information about the benefits of providing their child with support for PA [42].	Messages targeting parents should target parents affective and instrumental attitudes toward parent support for PA and focus on presenting the benefits of providing support for PA.

**Table 3 ijerph-18-07046-t003:** Literature regarding the dissemination of physical activity messages targeting parents.

Theme	Subtheme	Articles Identified	Main Relevant Findings	Recommendation for PA Message and Information Dissemination
(1) Dissemination to enhance cognitive processing	(a) Awareness	[10,42,44]	Message awareness is positively associated with favourable attitudes toward child PA among parents [42]. Compared to parents with low PA message awareness, parents with greater awareness were more likely to believe that PA offered benefits to their children [44]. Parents with greater PA campaign message awareness were more likely to believe their children needed to engage in more PA, had stronger intentions to provide parent support for PA, and exhibited greater support for PA compared to parents with low campaign awareness [10]. It is important to garner awareness to promote changes in beliefs and intentions toward child PA and parent support for PA [10,42,44].	Messages targeting parents should focus on dissemination strategies such as repeated exposure to promote campaign and message awareness which can positively impact pre-intentional factors.
	(b) Recall	[42,44,45,46]	PA message recall among parents is positively associated with attitudes, beliefs, and support for PA behaviours [42], as well as greater knowledge regarding child PA and increased family PA [45]. However, PA message recall is generally low among parents [44] unless it is prompted [46].	Messages targeting parents should focus on dissemination strategies that promote recall such as repeated exposure. Such strategies have the potential to positively impact pre-intentional factors.
(2) Social marketing strategies to enhance dissemination	(a) Marketing strategies	[47,48,49]	The success of a PA campaign in the United States called VERB is thought to be the result of the marketing professionals who developed the campaign based on extensive consumer research [47,48]. The VERB campaign was branded as cool and fun, and ensured that the PA messages reflected core attributes of the brand [48]. The VERB campaign employed social marketing tactics such as developing a brand affinity (i.e., messaging that aligned with the parents’ values) [49], using paid media advertising, contests and community-based events, as well as collaborating with popular celebrities or athletes [49].	Messages targeting parents should utilize marketing strategies for optimal dissemination such as audience research, channel placement and outcome evaluation.
	(b) Tailoring dissemin-ation for subgroups of parents	[48,49]	The VERB campaign successfully targeted parents of Asian, Indian, Latino, and African American backgrounds by tailoring PA messages to reflect dissemination needs and preferences of various subgroups (e.g., delivered in various languages, disseminated through preferred radio television stations [48,49]).	Messages targeting parents should tailor messages to subgroups of parents to enhance message dissemination to improve the uptake of PA messages.
	(c) Brand equity	[15,47]	To enhance brand equity, the VERB campaign’s use of popular television characters and airing messages during popular television times for parents and children [47]. Within this campaign, brand equity increased steadily over a two-year period [47]. Brand equity also increased among parents after six months of PA message exposure and parents who reported higher brand equity also reported higher levels of parent support for PA [15].	Messages targeting parents should utilize strategies to enhance brand equity (e.g., celebrity endorsement or credible messengers). Higher brand equity can lead to increases in parent support for PA or factors related to parent support for PA.
(3) Messages that target the dissemination preferences and suggestions of parents	(a) Preferred Message Dissemin-ation Approaches	[34,50,51,52,53,54,55]	Multi-platform approaches were highlighted in the current literature [34,50,52]. Parents preferred both digital and traditional forms of dissemination [34,53]. One study suggested the use of a mass media approach to reach a large population [53]. Two studies found text messages are both feasible and efficacious for disseminating information to parents [54,55]. Unique forms of PA message and information dissemination through storybooks or “parent nights” within the community may be novel and creative methods to motivate parent support for PA [50,51].	Messages targeting parents should utilize dissemination strategies that parents prefer. Such dissemination strategies should utilize a combination of both digital and traditional forms. Unique forms of dissemination can be used (e.g., text messages or parent nights) but in combination with other dissemination strategies.

**Table 4 ijerph-18-07046-t004:** Literature regarding the development of physical activity messages targeting parents of children with disabilities.

Theme	Subtheme	Articles Identified	Main Relevant Findings	Recommendation for PA Message and Information Development
(1) Common Barriers to Parent Support for PA Among Parents of CWD	(a) Lack of information	[17,56,57,58,59,60]	One content analyses of PA websites for CWD found that less than 25% of the websites provided accurate information or appropriate knowledge-based information (e.g., PA recommendations, definitions of PA, and barriers) [60]. A lack of access to accurate and disability-specific PA information can create feelings of frustration among parents of CWD [17,56,57,58,59] which can negatively influence motivation to provide support for PA. For example, mothers assessing the Canadian movement guidelines described them as “incompatible with the abilities, experiences, and needs of CWD” [57]. Parents of CWD have also expressed concern regarding the lack of clarity around the use of terms such as “inclusive” or “accessible” with PA messages [17,59]. Parents have expressed a need for inclusive images and modifications for certain behaviours within PA messages [57].	Messages targeting parents of CWD should focus on addressing the lack of disability specific information available to parents. Such messages can include information regarding clear and concise PA definitions and types of programming available, details regarding accessibility, inclusivity and support available, information regarding safety, and ideas for facilitating PA at home.
	(b) PA Barriers salient to parents of CWD	[17,56,57,58,59]	Commonly reported barriers included high costs of program participation [57], transportation coordination [57], lack of disability-specific PA opportunities [58], lack of targeted PA Information [17,58,59] and social inclusion [56]. The literature discussed the extraordinary efforts that are required for parents of CWD to support their child’s PA while overcoming barriers and balancing safety concerns with independence [17,56,59]. Heightened concerns regarding their children’s safety during PA participation is a prevalent issue for parents of CWD [56,59]. Parents have directly expressed a desire for information regarding the safety of PA opportunities and specific guidelines for children with varying disabilities [17,56,57].	Messages targeting parents of CWD should address common barriers that they experience (e.g., transportation coordination, cost of programs, safety, and social inclusion). Some suggestions include providing information regarding program staff, using inclusive images and providing safety information. Such messages can help parents feel prepared to overcome certain barriers they experience.
(2) Messages that target psychosocial antecedents of parent support for PA among parents of CWD	(a) Theoretical predictors of behaviour change	[60]	Literature highlights a lack of theory or evidence-based information with the PA website content targeting CWD [60]. Messages targeting self-efficacy were the most common source of messaging targeting a theoretical predictor of behaviour change, while less than 10% of web-content included messages targeting self-regulation, self-monitoring, and planning [60].	Messages targeting parents of CWD should target theory-based constructs such as pre-intentional factors (e.g., attitudes, perceived behavioural control and subjective norms) and post-intentional factors (e.g., behavioural regulation and planning) to enhance message effectiveness. Messages should facilitate planning and self-regulation regarding support for PA as such information is significantly lacking for parents of CWD specifically.

**Table 5 ijerph-18-07046-t005:** Literature regarding the dissemination of physical activity messages targeting parents of children with disabilities.

Theme	Subtheme	Articles Identified	Main Relevant Findings	Recommendation for PA Message and Information Dissemination
(1) Dissemination Preferences and Suggestions of Parents of CWD	(a) Preferred dissemination approaches	[17,58,59]	Parents of CWD can benefit from a multi-platform approach to message and information dissemination [17,58]. Parents of CWD expressed desire for information that is easily accessible and disseminated through credible and reliable sources [17,58,59]. Parents desire a “central hub” for finding targeted PA messages [17]. Many parents of CWD seek PA information and learn from other parents of CWD [17,58]. As such, communication spaces such as blogs, chat rooms, and message boards are of value to support parents in finding and sharing PA information [17].	Messages targeting parents should utilize dissemination strategies that parents of CWD prefer. Some suggestions include a “central hub” for information, information disseminated by credible sources, and a multi-platform approach.

## Data Availability

Data is contained within the article.

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
