# Peer review of "Developing and Disseminating Physical Activity Messages Targeting Parents: A Systematic Scoping Review"

_ijerph, 2021, doi:10.3390/ijerph18137046_

Round 1

Reviewer 1 Report

Please see the attached

Author Response

Please see the attached document. Responses to the reviewer are in red font. 

Reviewer 2 Report

The article is relevant given the implications of physical activity in the child population and the responsibility of parents in the face of the phenomenon of sedentary lifestyles, chronic non-communicable diseases and the vulnerability of the group of children with disabilities. It is a key contribution to developing and disseminating messages for decision making.

It meets most of the PRISMA-ScR checklist; however I have some observations.

  1. Form: lines 142 and 249 do not show the number of duplicate and excluded articles, although figures 1 and 2 do identify them.
  2. Adequately summarise the results because in the wording of the text and tables 3 and 4 where the identified themes and sub-themes are shown, the information is duplicated.
  3. I am concerned about the analysis of the quality and risk of bias in the review process and publication bias of the studies as it is not made explicit whether any tool or instrument was used for the critical appraisal of cross-sectional studies, it is not clear whether the risk of bias in the studies was low, moderate or high risk of bias, how many of them were in that category? It is said that there was a choice of studies by consensus when discrepancies were noted but was there any intra-observer analysis? Assessment criteria guidelines are not observed.

Author Response

(The authors gave the same response as above.)

Reviewer 3 Report

This study addresses an interesting but under-researched topic, physical activity messages, which is relevant to the field of health. 
More specifically, this study systematically reviewed the extant literature on the development and dissemination of optimally effective PA messages targeting parents. Given the importance of this topic, the systematic review on this topic can provide insightful implications for future research. 
However, this study did not provide a strong research gap and rationale for this study. 

Moreover, after searching the relevant literature, it was found that no systematic review has been conducted on this topic, indicating the originality and need for this study. This paper is very well-written and clearly organized. Especially in the methodology section, the data selection and collection process are very clear. 

In addition, this paper is easy to read and the results are very straightforward. It also should be noted that the interpretation and discussion of results are very comprehensive. The conclusions are consistent with the findings and arguments of this study. 

In sum, this is a very well-written study addressing an important but under-researched topic in the fields of physical activity and health. 

Author Response

(The authors gave the same response as above.)
